# New Insights on *Lilium* Phylogeny Based on a Comparative Phylogenomic Study Using Complete Plastome Sequences

**DOI:** 10.3390/plants8120547

**Published:** 2019-11-27

**Authors:** Hyoung Tae Kim, Ki-Byung Lim, Jung Sung Kim

**Affiliations:** 1Institute of Agricultural Science and Technology, Chungbuk National University, Chungbuk, 28644, Korea; rladbgus@gmail.com; 2Department of Horticulture, Kyungpook National University, Daegu, 41566, Korea; kblim@knu.ac.kr; 3Department of Forest Science, Chungbuk National University, Chungbuk, 28644, Korea

**Keywords:** *Lilium*, phylogenomics, plastome, molecular markers, gene tree, species tree

## Abstract

The genus *Lilium* L. is widely distributed in the cold and temperate regions of the Northern Hemisphere and is one of the most valuable plant groups in the world. Regarding the classification of the genus *Lilium*, Comber’s sectional classification, based on the natural characteristics, has been primarily used to recognize species and circumscribe the sections within the genus. Although molecular phylogenetic approaches have been attempted using different markers to elucidate their phylogenetic relationships, there still are unresolved clades within the genus. In this study, we constructed the species tree for the genus using 28 *Lilium* species plastomes, including three currently determined species (*L. candidum*, *L. formosanum*, and *L. leichtlinii* var. *maximowiczii*). We also sought to verify Comber’s classification and to evaluate all loci for phylogenetic molecular markers. Based on the results, the genus was divided into two major lineages, group A and B, consisting of eastern Asia + Europe species and Hengduan Mountains + North America species, respectively. Sectional relationships revealed that the ancestor *Martagon* diverged from *Sinomartagon* species and that *Pseudolirium* and *Leucolirion* are polyphyletic. Out of all loci in that *Lilium* plastome, *ycf1*, *trnF-ndhJ*, and *trnT-psbD* regions are suggested as evaluated markers with high coincidence with the species tree. We also discussed the biogeographical diversification and long-distance dispersal event of the genus.

## 1. Introduction

The genus *Lilium* L. is the type genus of Liliaceae that consists of approximately 100 species spread throughout the cold and temperate regions of the Northern Hemisphere [1,2]. *Lilium* species are economically important because of their ornamental features in horticulture as cut flowers and as potted and garden plants [3]. In addition, the flowers and bulbs of cultivated *Lilium* species are used for food and medicine [4].

The classification of the genus *Lilium* had been built based on flower shape before Comber’s classification [5]. As a result, the sectional (or subgeneral) boundaries in the genus frequently changed, and many species were referred to different sections according to different classification systems [6,7,8]. In contrast to the previous sectional concept based on flower shape, Comber [5] suggested the use of new natural characteristics, including characteristics of leaves, bulbs, and stems to classify the genus *Lilium* and divided it into seven sections. Although Comber’s classification was revised by De Jong [9] based on previous papers published up to that time, Comber’s classification has been primarily used to date to recognize *Lilium* species and circumscribe the sections within the phylogeny of *Lilium*.

Nishikawa et al. [10] constructed the phylogeny of 55 *Lilium* species using the internal transcribed spacer (ITS) region to clarify their phylogenetic relationships and found that most species formed a clade according to the classification based on the morphological features. Nonetheless, it is difficult to follow their results directly because most of the branch lengths were very short and not significantly supported. In addition, the phylogenies using the same marker with more samples indicated that most of the sections, except for *Martagon*, were polyphyletic [11,12,13]. Such discordance between classifications based on morphological characteristics and molecular phylogeny has also been found in other genera. For example, the subgenera of the genus *Cymbidium* based on morphological characteristics [14] was not found to be monophyletic using ITS [15] and ITS + *matK* [16,17], and a number of branches in the phylogenetic trees were collapsed in strict consensus trees. It is likely that gene trees based on molecular markers were often insufficient for investigating species trees because of certain evolutionary events, such as incomplete lineage sorting and horizontal gene transfer, or because convergent morphological evolution has occurred in certain lineages. To ensure the phylogenetic accuracy, increased taxa sampling has been preferred for a long time [18,19,20]. However, it has been suggested that the number of genes may be a more important determinant than the taxon number [21,22,23], and the rapid development of next-generation sequencing techniques has led us to the phylogenomic era, which provides numerous data to resolve ambiguous relationships.

Plants contain three genomes: a nuclear genome, a mitochondrial genome, and a plastid genome (plastome). The nuclear genomes [24] and mitochondrial genomes [23] in flowering plants substantially vary in length, whereas the plastomes maintain consistent lengths and typical structures for a long time [25], except for those of some specific lineages [26,27,28]. In addition, the typical plastome structure allows for next-generation sequencing (NGS) data to be assembled easily. As a result, the number of deposited plastome sequences is 10 times that of mitochondrial genome sequences in the NCBI genome database. The highly conservative nature of the plastome structure makes it possible for intergenic spaces to be used as molecular markers, as well as genes [29,30]. Using this feature, many researchers have attempted to increase phylogenetic accuracy for unresolved taxa using whole plastome sequences [31,32,33] and have also suggested new combinations of molecular markers for specific taxa [34,35,36,37,38]. Two studies on the phylogenomics of *Lilium* using plastome sequences have been published [4,39] to date. The phylogenetic relationships among the branches were well resolved with high support. However, taxon samplings were restricted because of the inability to use the data from both papers, which were published in the same year. Therefore, it was difficult to verify Comber’s classification based on well-resolved phylogeny. Du et al. [4] suggested molecular markers for the phylogeny of *Lilium* based on nucleotide diversity; however, the issues of polytomies and low supporting values still remained. Apparently, there was no comparison between species trees and gene trees to suggest molecular markers for the phylogenetic analysis. A gene tree can be easily acquired from each gene; however, it can conflict with the species tree [40] because not all genes have evolved in the same manner in the same lineage. Therefore, it is necessary to evaluate the reliability of any constructed species trees and gene trees and suggest molecular markers for the phylogeny of certain lineages.

In this study, three plastomes in *Lilium* were newly sequenced (1) to verify Comber’s classification and (2) to evaluate all loci of the *Lilium* plastome for phylogenetic molecular markers. Two taxa, *Lilium formosanum* Wallace and *Lilium candidum* L., were sampled to investigate the monophyly of *Leucolirion* and to determine the phylogenetic position of *Liriotypus*. *Lilium leichtlinii* var. *maximowiczii* (Regel) Baker was added because it was found to be closely or distantly related to *Lilium lancifolium* Thunb. in the previous studies [10,11,12,13]. The phylogenies of the genus *Lilium* that were built based on different methodologies using plastome sequences were constructed to determine a more accurate species tree. Based on this result, gene trees were compared to the species tree of the genus *Lilium* to evaluate which gene trees were similar to the species tree in a topology, while reflecting the generic relationships in Liliaceae. Additionally, based on the species tree generated in this study, the evolution of genomic characteristics in the genus *Lilium* was also discussed.

## 2. Results

### 2.1. Newly Sequenced Plastomes of Three Lilium Species

Numbers of reads ranging from 35,028,408 (*L. leichtlinii* var. *maximowiczii*) to 89,224,524 (*L. candidum*) were used for raw data after removing less than 50 bp reads from each dataset for the three species (Table 1). After constructing a complete plastome sequence, the number of mapped reads to complete the plastome sequences ranged from 568,878 (*L. leichtlinii* var. *maximowiczii*) to 1,945,534 (*L. formosanum*). Consequently, the average coverage of each plastome sequence ranged from 520.1 (*L. leichtlinii* var. *maximowiczii*) to 1840.6 (*L. formosanum*). The plastomes of the three *Lilium* species (*L. candidum*, *L. formosanum*, and *L. leichtlinii* var. *maximowiczii*) were 152,101–152,653 bp in length with a large single copy (81,481–82,101 bp), a small single copy (17,524–17,644 bp), and two inverted repeats (26,488–26,514 bp) (Appendix A).

The overall GC content was 37.0%, which is similar to those of other *Lilium* species. In total, 133 genes were annotated from each plastome with 85 protein-coding genes, 8 rRNA genes, 38 tRNA genes, and 2 partial genes (*rps19* and *ycf1*). *InfA* was a pseudogene in all three plastomes, as well as other *Lilium* species, and *cemA* was pseudogenized in the plastomes of *L. candidum* and *L. leichtlinii* var. *maximowiczii* because of the copy number variation of the poly A-tract.

### 2.2. Nucleotide Diversity within Genera Lilium and Fritillaria

The total length of the aligned sequences of *Lilium* and *Fritillaria* plastomes was 159,458 bp. The maximum nucleotide diversities of *Lilium* and *Fritillaria* were 0.030 and 0.041, respectively (Figure 1). In total, the nucleotide diversity was lower in the inverted repeat (IR) region than in the large single copy (LSC) and small single copy (SSC) regions in both genera, as well as being inversely proportional to the GC content. The delta nucleotide diversity between the two genera fluctuated from 0.014 to −0.017. Four loci higher than 0.02 in *Lilium* plastomes and two loci higher than 0.025 in *Fritillaria* plastomes were caused by large deletions in certain species because we deleted the sites with missing data for at least one species. In addition, there were the loci with higher nucleotide diversity, particularly in *Lilium* plastome sequences, which were caused by small palindromic repeats and the IR expansion from IRa to SSC.

### 2.3. Phylogeny of Lilium in Liliaceae

The 28 phylogenetic trees constructed from the four datasets, four tools, and two models showed that all of the genera in Liliaceae were monophyletic (Figure 2). *Amana* and *Erythronium* in tribe Tulipeae were distinguished from *Cardiocrinum*, *Fritillaria*, and *Lilium* in tribe Lilieae. In tribe Lilieae, *Cardiocrinum* diverged first, and *Lilium* and *Fritillaria* were separated later. In contrast to the generic relationships within the tribe, the infrageneric relationships varied slightly among phylogenetic trees.

In the clades of *Lilium* in 28 phylogenies, different datasets affected the change in topology more than different tools and models; however, the major clades were highly conserved in all phylogenies (Appendix A). The genus was divided into two major lineages, group A and group B, consisting of eastern Asia + Europe species and Hengduan Mountains + North America species, respectively (Figure 3). Group A consisted of three clades and five independent lineages. Clade I comprised three *Martagon* species and *L.* sp. KHK_2014, except for the phylogeny created by ASTRAL with coding genes. Clade II consisted of *L. amabile*, *L. lancifolium*, and *L. callosum*, which belong to *Sinomartagon*. *L. longiflorum,* and *L. formosanum* of *Leucolirion* and *L. brownii* of *Archelirion* formed clade III. *L. cernuum* of *Sinomartagon* and *L. candidum* of *Liriotypus* were sister groups to *Martagon* and the rest of group A, respectively. The positions of *L. leichtlinii* var. *maximowiczii* were distantly related to *L. lancifolium* and *L. amabile*, which were morphologically close to *L. leichtlinii* var. *maximowiczii*.

Group B consisted of four clades and two independent lineages. Clade IV comprised species in *Sinomartagon* 5c. Clade V consisted of *L. leucanthum* of *Leucolirion* and *L. henryi* of *Sinomartagon* 5a. Clade VI comprised *L. duchartrei* and *L. fargesii* of *Sinomartagon*, and clade VII included *L. washingtonianum*, *L. pardalinum*, and *L. superbum* of *Pseudolirium*. *L. distichum* of *Martagon* and *L. philadelphicum* of *Pseudolirium* were sisters to clade IV and clade IV + V, respectively. Most of the branches, including seven clades in groups A and B, were strongly supported by bootstrap values of the maximum likelihood analysis, branch support values of ASTRAL, and posterior probabilities of Bayesian inference, but certain branches within clades had moderate support.

A consensus tree of 28 phylogenies was constructed using the 50% majority rule to infer a robust species tree of *Lilium* (Figure 2). Except for two polytomies, one in *Lilium* and another in *Amana*, the seven clades in *Lilium,* as described above, were maintained.

### 2.4. Comparing Gene Trees to Species Trees

Among the 189 total gene trees, only 22 gene trees showed that each genus was monophyletic, even though two trees built using *trnS-trnG* IGS and the *trnV* intron had different generic relationships compared with the other 20 gene trees (Appendix A). Thirty-six clusters of similar trees from the 190 trees (189 gene trees + the consensus tree of 28 species trees) were generated using treespace [43], with five principal components and a cut-off height of 100 (Figure 4). Among them, 21 clusters consisted of more than three trees (Table 2), and the 10th cluster included a consensus tree of the 28 species trees and 21 gene trees. These 21 gene trees were identical to the gene trees wherein each genus was monophyletic, except for the *ycf2* gene tree.

### 2.5. IR Expansion and Contraction in Lilium

Based on the topology of *Lilium* species using Bayesian inference with all loci and partition models, the movements of IR boundaries (LSC-IRb, IRb-SSC, and SSC-IRa) were investigated (Figure 5). The LSC-IRb boundaries of group A were identical to each other. In contrast to group A, IR expansions and contractions were found in group B, excluding *L. distichum*, *L. washingtonianum*, *L. pardalinum*, and *L. superbum*. Two IR-SSC boundaries were more diverse than LSC-IRb, although most IR expansions and contractions occurred in the SSC-IRa boundary. Overall, IR boundaries were highly conserved within clades, except clades IV and VI.

### 2.6. Insertions/Deletions in Lilium

In total, eight insertions/deletions (indels) longer than 50 bp occurred in over five species that were found throughout the whole plastome sequences. Five of them had distinguishable features between groups A and B in the *Lilium* phylogeny (Appendix A), but others did not correspond to the phylogenetic relationships (Appendix A).

## 3. Discussion

### 3.1. Evolution of Lilium Plastomes in Liliaceae

The plastome sequences of *Lilium*, including the three newly determined species in this study, are highly conserved in terms of gene content and order and genomic structure. Although the nucleotide diversities of the LSC and SSC regions were higher than that of the IR region, this is a common phenomenon across the angiosperms [25]. However, the nucleotide diversities of *Lilium* and *Fritillaria* were higher in the LSC and SSC regions but lower in the IR region as compared to that of *Paris* belonging to Melanthiaceae of Liliales [44]. Consequently, the IR regions in Liliaceae appear to be more stable by purifying selection, and the LSC and SSC regions were assumed to have been under relaxed purifying selective pressure compared to those of Melanthiaceae during their evolutionary process. In Liliaceae, the fluctuating delta nucleotide diversity between *Lilium* and *Fritillaria* also supported the idea that different loci of the plastome have undergone different selective regions in this lineage, except the overestimated nucleotide diversities owing to deletions and small inversions. These findings imply that the mutational dynamics with respect to plastome loci between *Lilium* and *Fritillaria* have been processed differently even though they are closely related taxa.

### 3.2. Verification of Comber’s Sectional Classification

In this study, we reconstructed the phylogeny of 28 *Lilium* species using complete plastomes with four different datasets, four tools, and two models and compared these trees to determine the accurate species tree. There were slightly different topologies, but the major clades were coincident among the trees and were strongly supported by various branch support values (Appendix A). Before the discussion of the phylogeny of *Lilium*, we must evaluate the identification of plastome sequences of *L. distichum* (NC_029937) and *L.* sp. KHK_2014 (NC_027679), which were published by the same research group, because of their unreliable phylogenetic position in the genus. Du et al. [4] first raised a question regarding the phylogenetic position of *L. distichum* (NC_029937). The *rbcL* and *matK* sequences of *L. distichum* (NC_029937) were identical to those of *L. speciosum* (*rbcL*: AB034922.1; *matK*: AB030853, AB049526). On the contrary, *atpB*, *rbcL*, and *ndhF* of *L.* sp. KHK_2014 (NC_027679) were identical to those of *L. distichum* (*atpB*: KC796843, JX903928, KM085888; *rbcL*: JX903238, JN786059, JN417422, KP711933; *ndhF*: JX903509, KM085762). Consequently, it may be necessary to exclude these two plastomes or to consider *L.* sp. KHK_2014 as *L. distichum* to avoid improper conclusions for the phylogenomics of *Lilium*.

All phylogenetic trees using different datasets indicated that *Lilium* species could be divided into two major groups. These two groups were also distinguished by the mutational dynamics of the IR expansion/contraction and large indels (Figure 5 and Appendix AA).

#### 3.2.1. The Phylogenetic Position of Martagon

*Martagon* consists of five species that are primarily distributed in northeastern Asia and Russia, except *L. martagon*, which ranges widely from central Europe to eastern Siberia. This section had been considered an early-diverging lineage in *Lilium* based on the morphological characteristics: hypogeal and delayed germination, whorled leaves, jointed scales, and heavy seeds [5,45]. In contrast to the morphological analyses, molecular phylogenetic analyses using ITS sequences showed that this section is a more recently derived lineage and is sister to some of the *Sinomartagon* + *Leucolirion* [10,11,13]. Based on these subgeneric relationships, Gao et al. [41] suggested that the ancestor of *Martagon*, *Sinomartagon*, and *Leucolirion* 6b had a distribution within the Hengduan Mountains before *Martagon* separated from *Sinomartagon* + *Leucolirion* 6b approximately 8.8 million years ago. However, based on the plastome sequences, *Martagon* is not a sister to *Sinomartagon* + *Leucolirion* 6b but forms a clade within *Sinomartagon* I (Figure 5). Additionally, the plastome structures of *Lilium* provide further support. The junctions between SCs and IRs of *Martagon* are identical to those of *Sinomartagon* I, whereas they differ from those of *Sinomartagon* 5c and *Leucolirion* 6b (Figure 5). These results imply that the ancestor of *Martagon* diverged from *Sinomartagon* species, i.e., the ancestor of *L. cernuum* in this study, and the divergence time was more recent than the expectation of Gao et al. [41]. In addition, four species within *Sinomartagon* I and three species within *Martagon* are commonly distributed in eastern Asia. As a result, the hypothesis that the origin of *Martagon* is the Hengduan Mountains [41] is controvertible.

#### 3.2.2. The Polyphyly of Pseudolirium

*Pseudolirium* consists of all American lilies, including *L. philadelphicum*, which is a lectotype of the section [5]. Interestingly, when the data matrix involved *L. philadelphicum*, the phylogeny using ITS [10,11,13] showed the section was monophyletic, but using *matK* [46] revealed polyphyly for the section. Unfortunately, the phylogenetic position of the section using both markers was not resolved or was supported weekly. On the other hand, Kim et al. [47] suggested that *L. philadelphicum* seems to be distinguishable from other species in the section based on phylogenomics using plastome sequences. This relationship is well supported by the 28 species trees constructed in this study. In terms of DNA sequence mutations, a two base deletion in ITS was found specifically in *Pseudolirium* species, except for *L. philadelphicum* [10], and there was an obvious distinction between the IR-SC junctions of *L. philadelphicum* and other *Pseudolirium* species (Figure 5). Morphologically, subsection 2d, consisting of *L. philadelphicum* and *L. catesbaei* in *Pseudolirium*, is distinguished from other species by erect flowers and highly clawed perianth parts [3]. Consequently, a new circumscription of *Pseudolirium* should be considered to reflect the recent phylogenetic results.

#### 3.2.3. The Polyphyly of Leucolirion

*Leucolirion* consists of eight species with scattered and sessile leaves and trumpet flowers [1]. The section is subdivided into two subsections based on bulb color: dark purple or brown for 6a and white for 6b [1]. In this study, the two subsections were distantly separated with strong support and this result was congruent with the previous phylogenetic studies [13,46]. In addition, *L. henryi* of *Sinomartagon* 5a and *L. brownii* of *Archelirion* formed a robust clade with *Leucolirion* 6a and 6b, respectively (Appendix A, Figure 5). Based on the phylogenetic and cytological studies, Du et al. [13] suggested that *L. henryi* and *L. brownii* should be classified into *Leucolirion* 6a and 6b, respectively. Consequently, our results provide further support for the modification of *Leucolirion* according to Du et al. [13].

#### 3.2.4. The Position of Liriotypus in the Genus *Lilium*

*Liriotypus* comprises 20 species, including all European, Turkish, and Caucasian species, with the exception of *Lilium martagon* [5,48]. Among them, *L. bulbiferum*, having upright flowers, has been distinguished from the rest of the species within the section and forms a clade with the *Sinomartagon* species, including *L. dauricum* of *Daurolirion* based on the molecular phylogenetic analyses [13,48]. In this study, *L. bulbiferum* was placed far away from *L. candidum*, which is a lectotype of the section [5], agreeing with the results of previous studies. However, it forms a clade with *Martagon* + *Sinomartagon* I with strong support, although there are sampling gaps that prevent a concrete conclusion (Appendix A). Therefore, increased taxa sampling, particularly the members of *Daurolirion*, will help to resolve the discordance of the phylogenetic position of *L. bulbiferum* between this study and previous studies and offer the correct phylogenetic position for this species.

On the other hand, the phylogenetic position of *Liriotypus,* except *L. bulbiferum,* was irregular within the genus, although they formed a clade with the *Sinomartagon* 5c species-*Nomocharis* clade in previous studies [11,13]. On the contrary, *L. candidum* was an early-diverging taxon in group A, without alternative relationships with the rest of group A species in this study (Figure 3, Appendix A). Consequently, the newly suggested phylogenetic position of *Liriotypus* in this study is incongruent with that of previous phylogenetic studies using a few molecular markers. One possible explanation for this discordance is that the position of *L. candidum* does not represent *Liriotypus* in spite of its taxonomic importance by lectotype in the subsection. This species differs from other *Liriotypus* species based on rosette basal leaves and widely trumpet-shaped flowers [48,49], and the geographic circumscription of the sections in *Lilium* by Comber [5] collided with the molecular phylogeny of this study, i.e., *Pseudolirium* and *Liriotypus*. Another scenario is that the origin of *Liriotypus* came from early-diverging *Sinomartagon* (see additional details in the next section) and directly moved to Europe via the Caucasus during evolution.

In spite of low taxon sampling in *Liriotypus*, our result strongly supports the previous conclusions in which *L. bulbiferum* is separated from the rest of the species [48], and it suggests a new phylogenetic position of the section within the robust phylogenetic trees. However, because there still remains uncertainty as to whether the position of *L. candidum* belongs to the *Liriotypus* based on its unique morphological characteristics in the section, more sampling in the section will be needed to solve its position.

#### 3.2.5. Biogeographic History of the Genus *Lilium*

The geographical origin of the tribe Lilieae has been speculated to be the Himalayas + Hengduan Mountains and multiple intercontinental dispersal events for Lilieae were suggested, as well as *Lilium* [41,50]. In this study, we also found long distant dispersals and simpler than the previous speculation. Gao et al. [41] suggested a long-distance dispersal model of *Lilium* based on ITS and *matK* phylogenies, in which there were movements among four regions: Hengduan Mountains, eastern Asia, Europe, including Caucasus, and Northern America. However, from the results in this study, it was suggested that there were two main long dispersals in eastern Asia—Europe in group A and Hengduan Mountains—North America in group B (Figure 3).

To explain these distributions, it was supposed that *Lilium* comprised “*Sinomartagon* and its derivatives.” *Sinomartagon* comprises approximately 30 species [11] with epigeal and immediate germination (except *L. henryi*), scattered leaves, an entire bulb, and Turk’s cap flowers [5], and it has a distribution from the Hengduan Mountains to eastern Asia. In molecular phylogenetic trees, including those in this study, this section has polyphyly regardless of the types of marker or taxon samplings [12,13,41,46]. Therefore, if the manner for distant dispersals was paved during glacial periods or pollinators traveled a great distance at the beginning of the *Lilium* diversification, certain populations of different species in the *Sinomartagon* could have moved together. Therefore, the adaptations to new circumstances may have led to new populations that morphologically converged. Consequently, populations within new circumstances had similar morphological characteristics and belonged to the same section by morphological classification. This may cause the discordance between classifications based on morphological characteristics and molecular phylogeny. Regarding the basal lineages of group A and B, the “*Sinomartagon* and its derivatives” hypothesis is unclear because of our low taxon sampling or extinction of ancient *Sinomartagon* species. This hypothesis may further confuse the evolutionary history of the genus because (1) nobody has placed *Sinomartagon* as a basal lineage in the *Lilium* based on the morphological characteristics, and (2) most phylogenies of *Lilium* constructed using ITS are not consistent with the present results. In addition, inheritance from single parents, such as a plastome, makes it difficult to detect hybridization events. However, the phylogenetic tree generated in this study was the first robust phylogenetic tree with more than 20 samples. There is no doubt that the most important action for the discussion of phylogenetic relationships or divergence times of certain lineage is the construction of accurate species trees without polytomy and poor support. Therefore, we cannot rule out this “*Sinomartagon* and its derivatives” hypothesis based on the well-resolved phylogenetic tree.

### 3.3. Molecular Markers for the Phylogeny of Lilium and its Relatives

ITS has been used as a valuable molecular marker for the phylogeny of *Lilium* [10,11,13,41] but the phylogenetic relationships among sections or species were incongruent or were weakly supported. To overcome this problem, we constructed the phylogeny of *Lilium* using whole plastome sequences in this study. However, the production and manipulation of NGS data also require significant time and cost, as well as a higher-level technique than the Sanger sequencing method.

To evaluate which gene trees were similar to the species tree in terms of topology, while reflecting the generic relationship in Liliaceae, 189 gene trees were constructed, and only 20 were found to be consistent with the results presented in previous phylogenetic studies [51,52]. Among them, the *ycf1*, *trnF-ndhJ*, and *trnT-psbD* regions were coincident with highly variable regions of the *Lilium* plastome, as suggested by Du et al. [4].

This could serve as an alternative choice of NGS-based phylogenomics in *Lilium* when we use insufficient conditioned samples, such as very low concentration, fragmentation, or extraction from very old specimens that are difficult for use in the preparation of an NGS library.

## 4. Materials and Methods

### 4.1. Plant Materials and DNA Extraction

The bulbs of *L. leichtlinii* var. *maximowiczii* (Wooriseed, Korea) and the seeds of *L. formosanum* (Wageningen University, Netherlands) and *L. candidum* (Royal horticultural society Lily group, UK) were germinated on media at Kyungpook National University of Korea. Genomic DNA was extracted from young fresh leaves using a DNeasy plant mini kit (Qiagen, Valencia, CA, USA), following the manufacturer’s protocol.

### 4.2. Sequencing, Assembly, and Annotation

Genomic DNA was sequenced using the HiSeq 2500 instrument (Illumina, San Diego, CA, USA). The following assembling procedures were implemented using Geneious 10.2.5 [53]. Both ends of raw reads were trimmed with more than a 1% chance of an error per base. Reads exceeding 50 bp in length were extracted and used as raw reads after this step. Raw reads were mapped to the plastome sequence of *Lilium pardalinum* [47] with medium-low sensitivity. Reads were aligned to the reference then *de novo* assembled with zero mismatches and gaps among the reads to generate contigs. Raw reads were realigned to the contigs with zero mismatches and gaps among the reads for up to 100 iterations. The generated contigs were concatenated into a circular form using *de novo* assembled circularizing contigs with matching ends. Finally, the raw reads were mapped to the complete plastome sequence with zero mismatches and gaps among the reads to verify the coverage depths through the genome, because of the fact that many plastome-like sequences distributed in the mitochondrial genome and nuclear genome have relatively low coverage depths compared to that of the plastome.

All of the genes in the three plastome sequences were annotated and compared with those of *L. pardalinum* using Geneious annotation with 90% similarity, then re-checked separately using BLASTP [54] and tRNAscan-SE [55].

### 4.3. Sequence Diversity and GC Content Analyses

Twenty *Fritillaria* and 28 *Lilium* plastome sequences were aligned by MAFFT [56], then the AT-rich regions were realigned by MUSCLE [57] to increase the alignment accuracy at these regions. The alignment sequences were loaded in R ver. 3.5.1 [58], and the nucleotide diversities of the two genera were analyzed using sliding window analysis (window size = 600 bp, step size = 200 bp) by deleting the sites including at least one missing data point for all sequences. In addition, the GC content was also calculated to compare the relationship between nucleotide diversity and GC content.

### 4.4. Phylogenetic Analysis

Fifty-five plastome sequences from the genera *Lilium*, *Fritillaria*, *Cardiocrinum*, *Amana*, and *Erythronium* were downloaded from GenBank to construct a phylogeny for *Lilium*. We extracted 78 coding sequences (CDSs), 90 intergenic spaces (IGSs) longer than 100 bp, and 21 intron regions (two regions from *clpP*, *trnK*, and *ycf3*) from each plastome (Appendix A).

All of the extracted sequences were aligned using MAFFT [55] according to the region (Appendix A) and merged into four datasets as follows: All_loci (including CDSs, IGSs, and introns), CDS_loci, IGS_loci, and Intron_loci. The models for each partition for the four datasets were estimated using PartitionFinder 2 [59], ModelFinder [60], or Jmodeltest 2 [61] for different phylogenetic analyses. The phylogenetic trees were constructed using the RAxML Black Box with 1000 bootstraps [62] in the CIPRES gateway [63], MrBayes with ngen = 10,000,000, samplefreq = 1000, and burninfrac = 0.25 [64], or IQ-TREE with 1000 bootstraps [65]. In total, 189 gene trees were constructed using IQ-TREE with ModelFinder, and these were used to estimate the species trees by ASTRAL [66,67,68]. All of the details for phylogenies are summarized in Appendix A. A consensus tree of 28 phylogenies was constructed by majority rule.

### 4.5. Comparison of Gene Trees

The generic relationships of each gene tree were compared to those presented in previous phylogenetic studies of Liliaceae [51,52]. Clusters of similar gene trees were identified using the Kendall Colijn metric [69] in treespace [43] with five principal components and a cut-off distance of 100. Phylogeny using selected markers was constructed by IQ-TREE, with 1000 bootstraps for comparison to the consensus tree constructed by 28 phylogenies.

### 4.6. R Packages for Manipulation of Phylogenies

APE [70], Biostrings [71], dplyr [72], ggplot2 [73], ggtree [74,75], gridExtra [76], pegas [77], phytools [78], tidytree [79], and treespace [43] were used in this study.

## Figures and Tables

**Figure 1 plants-08-00547-f001:**
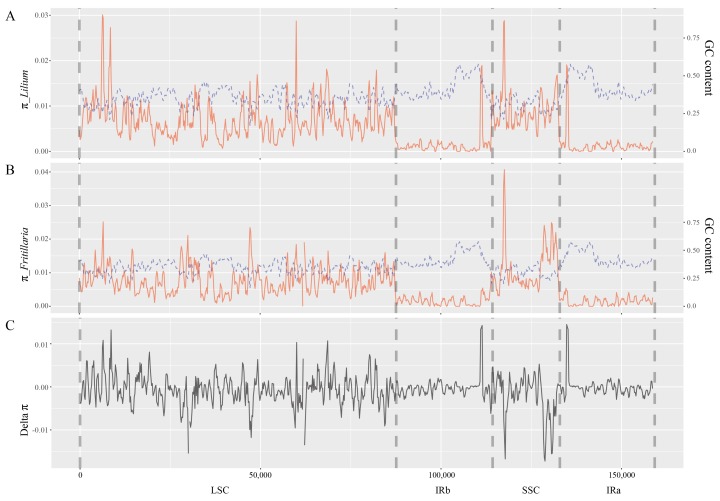
Nucleotide diversity (π) and GC content throughout the plastome sequence according to sliding window analysis (window size = 600 bp, step size = 200 bp). The red line and blue dashed line refer to π and GC content, respectively. The vertical dashed grey lines refer to the approximate boundaries of the plastome structure.

**Figure 2 plants-08-00547-f002:**
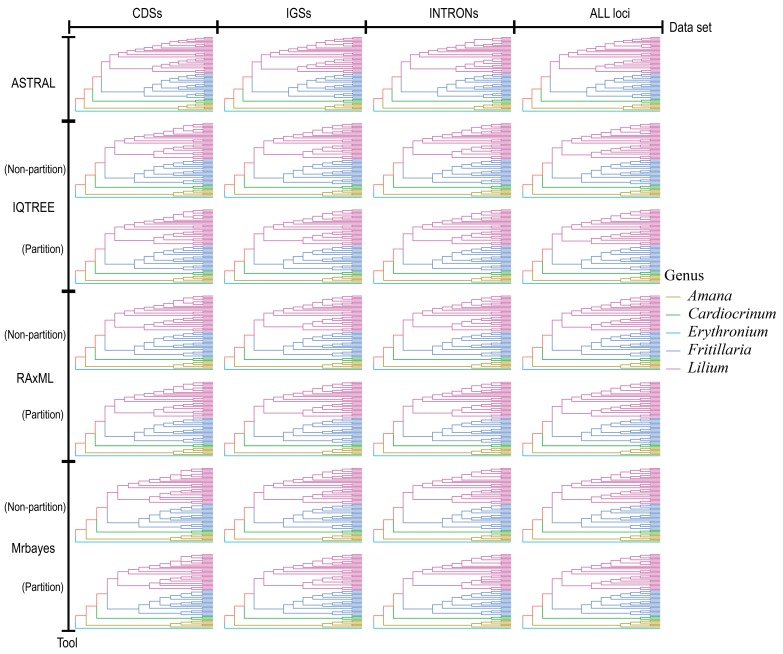
Phylogenetic trees based on four tools (ASTRAL, IQ-TREE, RAxML, and MrBayes) using four datasets (genes, introns, intergenic spacers, and all regions) and two different models (partition model and non-partition model of the dataset). The colored line refers to each genus.

**Figure 3 plants-08-00547-f003:**
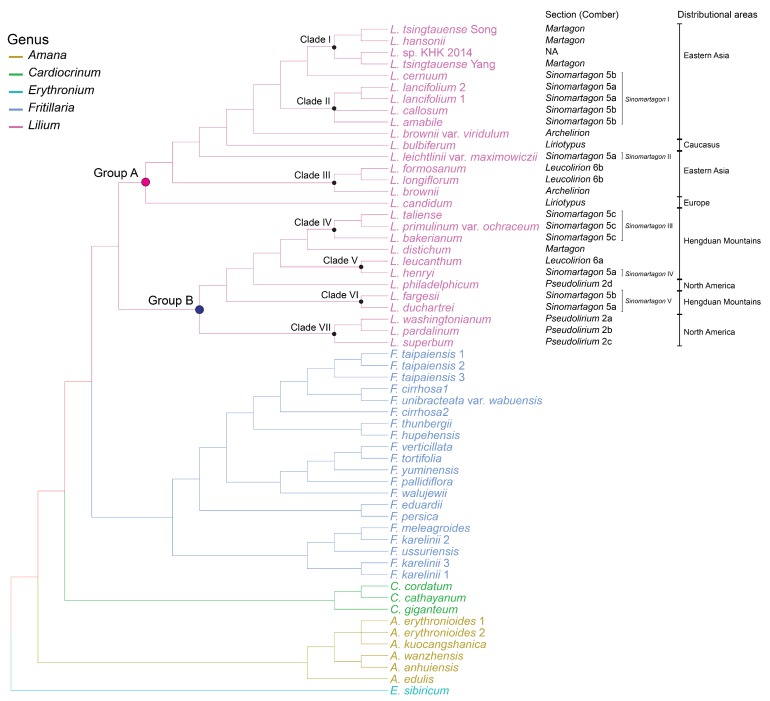
Consensus tree of 28 phylogenies based on four tools (ASTRAL, IQ-TREE, RAxML, and MrBayes) using four datasets (genes, introns, intergenic spacers, and all regions) and two different models (partition model and non-partition model of the dataset). The colored line refers to each genus. The distribution areas of clades are based on Gao et al. [41] and Xinqi et al. [42].

**Figure 4 plants-08-00547-f004:**
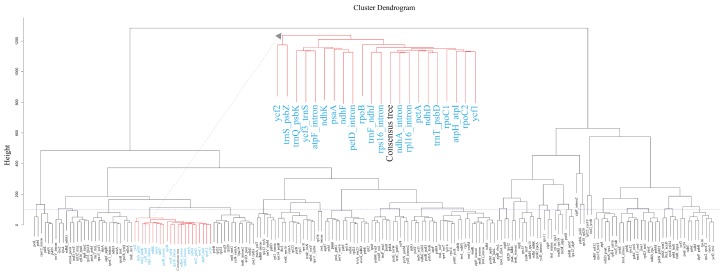
Hierarchical cluster analysis for 190 trees, including 189 gene trees and a consensus tree. The horizontal dashed line refers to the cut-off value for cluster trees.

**Figure 5 plants-08-00547-f005:**
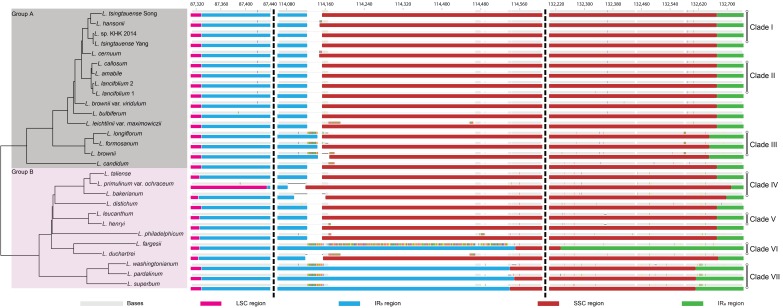
Inverted repeat (IR) expansion and contraction in *Lilium*. Pale grey refers to bases and colors on the bases represent disagreements with the consensus sequence. The red, blue, brown, and green blocks below bases stand for the large single copy (LSC) region, IRb region, small single copy (SSC) region, and IRa region, respectively. The tree on the left is constructed using Bayesian inference with all loci and partition models.

**Table 1 plants-08-00547-t001:** Summary of genome assembly.

Taxon	No. of Raw Reads (≥50 bp)	No. of Mapped Reads	Average Coverage	SRA ^a^ Accession
*Lilium candidum*	89,224,524	1,187,028	1134.7	SRR7617960, SRR7617961
*Lilium leichtlinii* var. *maximowiczii*	35,028,408	568,878	520.1	SRR7617965
*Lilium formosanum*	80,985,606	1,945,534	1840.6	SRR7617963, SRR7617964

^a^ Sequence Read Archive.

**Table 2 plants-08-00547-t002:** Information on 21 clustered trees using treespace, with five principal components and 100 cut-off distance.

Cluster	Clustered Trees	No. of Trees
1	*accD; ndhG; psbB; rpl20; rpl32; rpl33; rps19; accD_psaI; psaJ_rpl33; trnS_trnG; ycf4_cemA; trnK_intron2*	12
2	*atpA; ccsA; ndhA; ndhH; atpF_atpH; petA_psbJ; rpoB_trnC; rps15_ycf1; rps16_trnQ; trnE_trnT; trnT_trnL; rpoC1_intron*	12
3	*atpE; rps15; ndhH_rps15; ndhJ_ndhK; trnW_trnP*	5
4	*atpF; petG; psbH; psbT; ndhG_ndhI; petB_petD; psbA_trnK; trnN_ycf1*	8
5	*atpH; ndhC; psaI; psbE; psbF; psbJ; rpl2; rpl16; rpl23; ccsA_ndhD; petL_petG; trnA_rrn23; trnV_rrn16*	13
6	*atpI; ndhD_psaC; ndhF_rpl32; rpl14_rpl16; rps19_trnH; rrn5_trnR; trnS_rps4*	7
7	*cemA; petB; rbcL; rpl22; rps3; rps18; ndhC_trnV; rps11_rpl36; rps14_psaB; trnK_rps16; trnR_trnN*	11
8	*clpP; rps11; trnI_intron*	3
9	*matK; ndhB; ndhI; rpl14; rps7; rps14; psbE_petL; psbH_petB; psbJ_psbL; rps4_trnT; trnG_trnfM; trnL_ndhB; trnL_trnF; clpP_intron1; petB_intron; trnG_intron*	16
10	*ndhD; ndhF; ndhK; petA; psaA; rpoB; rpoC1; rpoC2; ycf1; ycf2; atpH_atpI; trnF_ndhJ; trnQ_psbK; trnS_psbZ; trnT_psbD; ycf3_trnS; atpF_intron; ndhA_intron; petD_intron; rpl16_intron; rps16_intron;* Consensus tree of species trees	22
11	*ndhE; ndhJ; psbC; rpoA; rps16; clpP_psbB; rps2_rpoC2; ycf2_trnL*	8
12	*petN; psbL; psbN; rps12_intron*	4
13	*psaB; psbA; rps8; psbM_trnD; rrn16_trnI; trnP_psaJ*	6
14	*psaC; psaJ; psbM; ndhB_rps7; psbN_psbH; rpl23_trnI*	6
15	*psbD; psaA_ycf3; rps12_trnV; trnV_trnM; ndhB_intron; rpl2_intron*	6
16	*psbK; psbZ; rps12; psbC_trnS; rrn4.5_rrn5; trnA_intron*	6
17	*rps2; rps4; trnV_intron*	3
18	*ycf4; petG_trnW; petN_psbM; trnC_petN; ycf3_intron2*	5
19	*atpI_rps2; cemA_petA; ndhE_ndhG; rpl32_trnL; rpl33_rps18; rpl36_rps8; trnD_trnY*	7
20	*psbI_trnS; rps8_rpl14; trnR_atpA*	3
21	*rpl16_rps3; trnfM_rps14; trnM_atpE*	3

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
