# Peer review of "New Insights on Lilium Phylogeny Based on a Comparative Phylogenomic Study Using Complete Plastome Sequences"

_plants, 2019, doi:10.3390/plants8120547_

Round 1

Reviewer 1 Report

The present experiment conducted a comparative phylogenomic study for Lilium using complete plastome sequences. A well-written manuscript will add important information and assist further research

Some minor modification has given below

The abstract should contain a summary of the results and should exclude an introduction. The abstract must provide enough information that will allow a reader to understand the key findings of the research. Supplementary figures should be presented in PPT or another format in a separate file. It is unreadable now.

Author Response

Thank you for your comment for improving our manuscript.

We revised the manuscript based on your comment as below.

The abstract should contain a summary of the results and should exclude an introduction.

The abstract must provide enough information that will allow a reader to understand the key findings of the research.

→ We rechecked the abstract and slightly revised to explain more clearly the result.

Supplementary figures should be presented in PPT or another format in a separate file. It is unreadable now. 

→ We prepared the new version of supplementary file which can be uploaded on the system.

Reviewer 2 Report

The manuscript “New Insights on Lilium phylogeny based on a comparative phylogenomic study using complete plastome sequences” by Kim et al. uses whole plastome sequences from 28 Lilium species to resolve phylogenetic questions in the genus. Even if most of the chloroplasts have been acquired from previous studies and only three are produced for this study, the study brings novel information that help to understand the Lilium phylogeny. As the authors mention in the introduction, the previous studies that produced the majority of the chloroplast sequences were published in the same year and both included a limited taxon sampling. To combine these sequences gives a better overview of the entire genus. In addition, the three new chloroplast sequences provide important genomic resources for the genus.

The manuscript has interest especially for research on Lilium and related species, but the analysis on marker congruency brings a wider perspective to the study.
In general, the manuscript is clear and written in good English. The phylogenetic analyses are thorough and include comparisons of different methods and data sets.

Minor comments
Line 17: It is not clear from the context if “newly determined species” refers to recently determined species or species determined in the current manuscript.
Line 47: Change “significantly positive” to “significantly supported” or “statistically supported” or something similar.
Lines 70-71: The first referred citation in “Two studies on the phylogenomics of Lilium using plastome sequences have been published [5,39] to date.” is probably wrong, maybe it should be number 4?
Lines 331-334: The concluding remarks of this paragraph are not clear to me. What is the hypothesis?

Author Response

Thank you for your comment for improving our manuscript.

We revised the manuscript based on your comment as below.

Line 17: It is not clear from the context if “newly determined species” refers to recently determined species or species determined in the current manuscript. 

→ We changed the text to “currently determined specie”.

Line 47: Change “significantly positive” to “significantly supported” or “statistically supported” or something similar. 

→ We changed the text to “significantly supported”.

Lines 70-71: The first referred citation in “Two studies on the phylogenomics of Lilium using plastome sequences have been published [5,39] to date.” is probably wrong, maybe it should be number 4? 

→ It was a mistake. The references were correctly cited to 4 and 39.

Lines 331-334: The concluding remarks of this paragraph are not clear to me. What is the hypothesis?

→We exclude some expression from the text because it bothers to understand the sentence clearly. And we add the “Sinomartagon and its derivatives” in front of “hypothesis” in the last sentence. We would like to emphasize the importance of the accurate phylogeny like our result to discuss the evolutionary history.

Reviewer 3 Report

The results obtained in the manuscript entitled "New insights on Lilium phylogeny based on a comparative phylogenomic study using complete plastome sequences" are reliable and properly presented.

Author Response

The results obtained in the manuscript entitled "New insights on Lilium phylogeny based on a comparative phylogenomic study using complete plastome sequences" are reliable and properly presented.

→ Thanks you for your kind response about our manuscript.